# Chromosomally Unstable Gastric Cancers Overexpressing Claudin-6 Disclose Cross-Talk between HNF1A and HNF4A, and Upregulated Cholesterol Metabolism

**DOI:** 10.3390/ijms232213977

**Published:** 2022-11-12

**Authors:** Sanyog Dwivedi, Georgina Hernández-Montes, Luis Felipe Montaño, Erika Patricia Rendón-Huerta

**Affiliations:** 1Laboratorio de Inmunobiología, Departamento de Biología Celular y Tisular, Facultad de Medicina, UNAM, Ciudad de México 04510, Mexico; 2Red de Apoyo a la Investigación, Instituto Nacional de Ciencias Médicas y Nutrición “Dr. Salvador Zubiran”, Ciudad de México 14080, Mexico

**Keywords:** gastric cancer, claudin-6, cholesterol metabolism

## Abstract

(1) Abnormally increased expression of claudin-6 in gastric cancer is considered a prognostic marker of the chromosomal unstable molecular subtype. However, a detailed molecular profile analysis of differentially expressed genes and affected pathways associated with claudin-6 increased (Cldn6^high^) expression has not been assessed. (2) The TCGA Stomach Adenocarcinoma Pan-Cancer Atlas Data was evaluated using Cytoscape’s Gene Mania, MCODE, and Cytohubba bioinformatic software. (3) 96.88% of Cldn6^high^ gastric cancer tumors belonging to the chromosomal unstable molecular subtype are associated with a worse prognosis. Cldn6expression coincided with higher mutations in TP53, MIEN1, STARD3, PGAP3, and CCNE1 genes compared to Cldn6^low^ expression. In Cldn6^high^ cancers, 1316 genes were highly expressed. Cholesterol metabolism was the most affected pathway as APOA1, APOA2, APOH, APOC2, APOC3, APOB-100, LDL receptor-related protein 1/2, Sterol O-acyltransferase, STARD3, MAGEA-2, -3, -4, -6, -9B, and -12 genes were overexpressed in Cldn6^high^ gastric cancers; interestingly, APOA2 and MAGEA9b were identified as top hub genes. Functional enrichment of DEGs linked HNF-4α and HNF-1α genes as highly expressed in Cldn6^high^ gastric cancer. (4) Our results suggest that APOA2 and MAGEA9b could be considered as prognostic markers for Cldn6^high^ gastric cancers.

## 1. Introduction

Gastric cancer (GC) is the sixth most common cancer and its global mortality (700,000 deaths) and morbidity (1.8 million cases) rates in 2020 is extremely high [1]. As GC is considered a clonal disease, genetic and nongenetic variations contribute to tumor heterogeneity, therefore a vast array of classification systems has been proposed [2]. The Cancer Genome Atlas Research Network evaluated molecular profiles of gastric adenocarcinomas and proposed four different subtypes: Epstein–Barr virus positive tumors (EBV), microsatellite unstable tumors (MSI), genomically stable tumors (GS), and chromosomal instable tumors (CIN) [3]. Specific genetic and epigenetic alterations of tumor suppressors and oncogenes among the molecular subtypes helped in the identification of novel prognostic and therapeutic targets. However, the existence of subclonal sets of the subtypes compels further investigation for the identification of targeted therapies.

Claudin-6 (Cldn6) is a tight junction protein expressed in embryonic and fetal gastric tissue, but not in healthy adult tissue [4,5]. Dysregulated expression of Cldn6 has been reported in gastric cancer [6] and it has been proposed as a single prognostic marker and tumor-promoting gene of a subset of genomically stable tumors [7]. The latter, together with the microsatellite stable/epithelial to mesenchymal transition phenotype subtype, has worse disease-free survival than other subtypes [8]. The human-derived gastric adenocarcinoma cell line AGS cells transfected with claudin-6 have increased proliferation, cell migration, and invasiveness [9]. Claudin-6 promotes tumor progression through the YAP1-snail1 axis affecting the transcription of cellular communication network target genes [10]. Though the importance of Cldn6 has been established, there is contradictory evidence of its expression and role [11,12] but, most importantly, enhanced expression of Cldn6 is not widespread in gastric cancer and its association with specific molecular pathways and malignancy is not yet completely understood.

Microarray and advanced sequencing technologies has speeded the search for biomarkers of diagnosis, treatment, and prognosis [13]. The application of diverse and enhanced bioinformatics tools that integrate multidimensional genomic characteristics have revealed new and valuable information [14,15,16]. The aim of this study was to analyze TCGA-Stomach Adenocarcinoma (STAD) data for tumors with enhanced Cldn6 (Cldn6^high^) expression using several bioinformatic tools to define whether its expression is associated with a more aggressive phenotype. We identified key pathways and driver genes that can be used as novel prognostic markers and identified genes with very high connectivity in the genetic interaction network, the so-called hub genes, and transcription factors that can become therapeutic targets for Cldn6 expressing gastric cancers.

## 2. Results

### 2.1. Cldn6 Expression and Molecular Subtype

The results of the TCGA GC data showed that 69 of the 71 Cldn6^high^ tumor samples belonged to the chromosomal instable molecular subtype, whereas only two belonged to the genomically stable subtype, however, the Cldn6^low^ tumor samples were distributed variably into chromosomal instable (41.27%), microsatellite unstable (27.78), Epstein–Barr virus positive (15.8%), genomically stable (13.49%), and POLE (2.38%) (Table 1). Survival plots showed a worsening trend in prognosis and progression-free survival for the Cldn6^high^ subset than the Cldn6^low^ group (Figure 1). 

### 2.2. Genomic Alterations and Differential Expression of Genes 

The Cldn6^high^ subset showed a higher prevalence of genomic alterations in TP53, MIEN1, STARD3, PGAP3, and CCNE1 (Table 2). The major mutations in TP53 included missense, truncating, and splice mutations (Appendix A). MIEN1, STARD3, PGAP3, and CCNE1 showed a tendency of amplification and co-occurrence in both subsets, but the frequency was higher in the Cldn6^high^ subset (Appendix A). Genomic alterations in AIRD1A and DOCK3 were predominant in the Cldn6^low^ subset (Table 2). In parallel with genomic alterations MIEN1, STARD3, PGAP3, and CCNE1 showed significantly higher (*p* < 0.004) mRNA levels in the Cldn6^high^ subset, whereas AIRD1A and DOCK3 mRNA levels do not endure differential expression among the subsets, TP53 showed a slightly lower mRNA level in the Cldn6^high^ subset compared to Cldn6^low^ subset (Figure 2a). As MIEN1, STARD3, and PGAP3 are part of the ERBB2 co-amplification gene group [17], the results revealed a significantly higher ERBB2 expression (*p* < 0.0001) in the Cldn6^high^ subset (Figure 2b). 

Out of the 18,881 genes evaluated, the differential expression of genes in Cldn6^high^ over Cldn6^low^ subsets showed that 1316 genes were overexpressed (log ratio ≥ 1) in the former subset whereas 101 genes were downregulated (log ratio ≤ −1) (Figure 3a). Melanoma antigen gene A-2, -3, -4, -6, -12, and -9B genes, as well as IGF2BP1, APOA1, and CSAG, were among the top 10 overexpressed genes (Table 3). These DEGs were expressed significantly higher in the Cldn6^high^ subset (Figure 3b). The results of the survival analysis revealed that the overexpression of these genes leads to a worse prognosis (Figure 3c). 

### 2.3. GSEA and Pathway Analysis 

The gene set enrichment analysis (GSEA) revealed upregulation of cholesterol metabolism, protein digestion, absorption, and steroid biosynthesis, and downregulation of pathways related to immunity and inflammation such as Th1 and Th2 differentiation, toll-like receptor signaling pathway, and B cell receptor signaling pathway in the Cldn6^high^ subset (Figure 4a). Cholesterol metabolism is the most importantly upregulated pathway in the Cldn6^high^ subset followed by vitamin and fat digestion and absorption (Figure 4b). All three pathways were interdependent and were supplemented by upregulated key apolipoproteins such as APOA1, APOA2, APOH, and APOB-100 (Figure 5). It is important to note that other cholesterol metabolism-related genes such as low-density lipoprotein receptor-related protein 1/2 (LRP1/2), sterol O-acyltransferase (SOAT1), and STARD3 genes were also upregulated (Figure 5). 

### 2.4. Hub Genes and Prognostic Markers 

Gene interaction analysis using genes with a log ratio ≥ 2 demonstrated several clusters, but only two with the highest scores were considered for the identification of hub genes. Cluster 1 with a score of 54.44 and cluster 2 with a score of 8.3 (Figure 6a,b). The top 10 hub genes with the highest interaction scores in cluster 1 are APOs (APOA1, APOA2, APOH, APOC2, APOC3), SERPINA7, SERPINC1, F9, PLG, and ITIH2 genes (Figure 6c). The top 10 hub genes with the highest interaction scores in cluster 2 are MAGEA genes (MAGEA9B, MAGEA2, MAGEC2, MAGEA4, MAGEA11, MAGEA6, MAGEA1, MAGEB2, MAGEA12, and MAGEA10) (Figure 6d). APOA2 and MAGEA9B had the highest degree scores in cluster 1 and 2, respectively. All these cluster 1 and 2 hub genes express highly and demonstrate the highest alteration frequency in the Cldn6^high^ subset (Figure 7a–d), but most importantly, the results showed that the presence of APOA2 and MAGEA9B in tumors is associated with a very low 10-year survival prognosis (Figure 8a,b). The KEGG analysis of cluster 1 hub genes showed that their major involvement is mainly in complement and coagulation cascades, cholesterol metabolism, and PPAR-related signaling pathways (Figure 9a,b). KEGG analysis was performed in cluster 2 hub genes, but a significant enrichment associated with a particular pathway was not established.

### 2.5. HNF-4α/HNF-1α

Statistical enrichment of cluster 1 DEGs to evaluate transcription factors enrichment showed that out of the thirteen, HNF-4α and PPARA were the most active in the Cldn6^high^ subset as opposed to RUNX3 and TBX21, which along with TP53 were also suppressed (Figure 10a). HNF-4α is responsible for the expression of 24.9% of the differential expressed genes, followed by RREB1 (20.3%), TCF3 (18.6%), and PPARG (17.3%) (Figure 10b). The expression of HNF-4α was significantly higher in the Cldn6^high^ subset than in the Cldn6^low^ subset (*p* < 0.0002) as determined by the mRNAz score (Figure 10c). An equivalent enrichment analysis of the top 10 genes included in cluster 1 showed that HNF-1α is responsible for the expression of 41.2% of the differential expressed genes, followed by HNF-4α (35.3%), and PPARG (23.5) (Figure 11a). The expression of HNF-1α was significantly higher in the Cldn6^high^ subset than in the Cldn6^low^ subset (*p* < 0.0001) as determined by the mRNAz score (Figure 11b) and its presence was associated to a low 10 year overall survival prognosis (Figure 11c). 

## 3. Discussion

Gastric cancer is a heterogeneous disease with vast phenotypic diversity that has been subjected to a profound revision as the traditionally histopathological classification of intestinal and diffuse types [18] has been enriched by molecular parameters due to the recognition of high-frequency gene mutations [19]. Analytical approaches have identified four subtypes of gastric cancer that provide a roadmap for patient stratification [3]. Each subtype is found throughout the stomach but differs based on sex, anatomic location, and clinical outcome [8,20]. The genomically stable subtype shares histological patterns with the diffuse type whereas the Epstein–Barr virus positive, microsatellite unstable, and chromosomal instable tumors are included in low- and high-grade dysplasia as well as intestinal or diffuse gastric cancer [21]. Since proteins carry out functions of genes and determine phenotype, proteomic profiling has identified signatures associated with the epithelial–mesenchymal transition of gastric epithelium, early-onset gastric cancer and tumor progression [22,23,24]. An important defensive mechanism of the gastric mucosa is the mucosal barrier composed of epithelial cells and intracellular junctions, especially claudins, which are their most essential tight junction’s component [25,26]. Amongst many, abnormal expression of claudin-6 has been recognized as a major promoter of the proliferation and migration of gastric cancer cells [10,27,28]. Therefore, we explored the proteogenomics changes of gastric cancer tumors associated with claudin-6 expression.

We observed that 97% of the Cldn-6^high^ tumors belong to the chromosomal instable subtype and that they are associated with a worse clinical outcome. The chromosomal instable subtype is characterized by mutations in the tumor suppressor gene TP53 [29]. Initially, it was suggested that TP53 mutations occur at a late stage of carcinogenesis [30], but it has turned out that this mutation might be a founding event in gastric cancer carcinogenesis [31]; however, the activity of P53 seems to be regulated by tight junctions [32] so the overexpression of claudin-6 could be associated with higher malignancy and consequently bad prognosis. In line with this possibility, our analysis showed that this Cldn6^high^ chromosomal instable subtype coincided with higher mutations in MIEN1, STARD3, and PGAP3 genes. MIEN1, migration and invasion enhancer 1, is involved in the negative regulation of the apoptotic process, positive regulation of cell migration, and filopodium assembly [33]. STARD3 overexpression results in increased aggressiveness by increasing membrane cholesterol and enhancing oncogenic signaling [34]. PGAP3 has also been involved in tumorigenesis and cancer progression [35]. These three genes are considered cancer-promoting genes co-amplified with the oncogene ERBB2 in gastric cancer [36]. Therefore, our results strongly suggest that overexpression of claudin-6 is strongly associated with enhanced tumor aggressiveness and invasiveness promotion.

Differential gene analysis demonstrated that several type I melanoma antigen gene (MAGE) A subfamily members were among the top 10 overexpressed genes in the Cldn6^high^ tumors, independently of their molecular subtype classification. Several members of the MAGE protein family are expressed in tumor cells; up-regulation of MAGE triggers the degradation of tumor suppressor P53, promoting tumorigenesis and aggressive tumor growth [37]. Type I MAGE expression has been related to low survival rates in advanced gastric cancer [38]. Our analysis pointed to MAGEA9B as being considered a hub gene. Overexpression of MAGEA9 correlates with unfavorable survival in lung adenocarcinoma [39], renal cell carcinoma [40], and esophageal adenocarcinomas [41], amongst many other tissues’ cancer. MAGEA9 has been speculated to function as an oncoprotein and to favor tumor cell survival [42]. Another overexpressed gene was CSAG, also known as TRAG-3, which has been found in 10% of gastric cancers, and most importantly, correlates with MAGE expression [43]; this correlation is associated with a poor prognosis [44]. Finally, the oncofetal IGF2BP1 gene that promotes tumor expression [45] was also overexpressed. Overall, the results suggest that claudin-6 expression, a protein mainly expressed in fetal tissues dedifferentiated cells [4,46,47] and strongly linked to the development of cancer [27], co-associates with genes involved in tumor-suppressing mechanisms. Claudin-6 has been proposed as a tumor suppressor gene in breast cancer [48,49].

The analysis of pathways associated with Cldn6^high^ tumors showed that many of the exchangeable apolipoproteins [50] gene families were overexpressed and that the cholesterol metabolism pathway was the most affected. Upregulation of these cholesterol genes ensures rewired cholesterol metabolism in Cldn6^high^ tumors and supports cancer progression. *APOA1* was among the top 10 overexpressed genes. It is well recognized that tumor microenvironments are exposed to metabolic challenges and that tumor cells have an increased demand for nutrients to support survival, especially in rapidly proliferating cells [51,52]. Interestingly, our analysis of transcription factors-related genes showed that HNF-1α and -4α were the most highly expressed in Cldn6^high^ tumors and Cldn6 requires both to be transcriptionally induced in renal epithelia [53]. It has recently been shown that the HNF1α-antisense RNA 1 is significantly upregulated in gastric cancer. HNF-4α plays a pivotal role in cholesterol metabolism [54], but it is also involved in the proliferation, invasion, and migration of cancer cells [55]. Overall, it is essential to determine if enhanced cholesterol metabolism in Cldn6^high^ tumors is an outcome of upregulated transcription factors, or it is only a part of the “building blocks” that sustain cell division [56].

The search for hub genes in tumor transcriptome has enabled the classification of human tumors into six different vascular signatures based on vascular “hub” genes [57]. We believe that our study provides an entry point for the identification of new markers and regulatory pathways in claudin-6 expressing gastric tumors.

## 4. Materials and Methods

### 4.1. Database

TCGA STAD Pan-Cancer Atlas gene expression data was obtained and analyzed for Claudin-6 expression using the CBioPortal [58,59]. Gastric cancer patient samples were grouped according to their Cldn6 expression into: Cldn6^high^ (n = 71) with Cldn6 mRNA z scores ≥ 5 and Cldn6^low^ (n = 138) with Cldn6 mRNA z scores ≤ 1. A Chi-square test was used to determine the molecular subtype with a significance *p*-value cutoff < 0.05 using the group comparison program of CBioPortal. 

### 4.2. Survival Analysis 

Survival data downloaded from the CBioPortal was analyzed using Kaplan–Meier curves with R (version 4.1.3) with survival v3.3.1; the output graphs were generated with survminer R v 0.4.9 packages considering time and status variables split by group. Survival analysis of upregulated genes (log ratio cutoff ≥ 2) was done using GEPIA2 (GEPIA 2 (cancer-pku.cn)) by parameters- group cut off at median, cutoff-high (%) and low (%) at 50% and 95% confidence interval. Survival estimates of individual genes were carried out using KM plotter (Kaplan–Meier plotter (Gastric) (kmplot.com)) with default settings.

### 4.3. Genomic Alterations and Differentially Expressed Genes 

Genomic alterations in both groups were analyzed with default settings of CBioPortal group comparison for overlap among patients/samples, with log-ratio calculated from Log2 based ratio of pct in (A) Cldn6^high^ group/pct in (B) Cldn6^low^ group, the *p*-value was derived from a one-sided Fisher exact test and q value was based on Benjamini–Hochberg procedure. Genes with Log ratio > 0 were enriched in Cldn6^high^ and genes with Log ratio <= 0 enriched in Cldn6^low^ tumors. Genomic alterations with q-Value < 0.05 were considered significant. Changes in gene expression were compared through mRNA expression levels of altered genes in both groups in terms of µ score (mean log2 expression of genes), σ score (standard deviation of log2 expression of genes), log-ratio (Log2 of the ratio of (unlogged) mean in the Cldn6^high^ group to (unlogged) mean in the Cldn6^low^ group), with *p*-value (derived from the Student’s *t*-test) and q-value (derived from Benjamini–Hochberg procedure). Volcano plots were generated in R (version 4.1.3) with ggplot2 and plotly R packages using an FDR < 0.05 and an absolute value of LogFC ≥ 1. Differential expression of genes was analyzed with GraphPad prism 9.0, and Statistical significance was confirmed with unpaired *t*-test and one-way ANOVA, *p*-value < 0.05 was considered significant. 

### 4.4. Signal Pathway and Transcription Factor Activity

Signaling pathway activities and transcription factor activities for the Cldn6^high^ group were determined with PROGENy v 1.18.0 and decoupleR v 2.2.2 to infer regulator activities with the weighted mean method with default settings.

### 4.5. Gene Set Enrichment Analysis (GSEA) and Gene Ontology (GO) Enrichment

GSEA of all DEGs with log ration cutoff ± 1 and with *p*-value ≤ 0.05 was done using WebGestalt (WEB-based Gene SeT AnaLysis Toolkit) with FDR q value ≤ 0.1. Gene cluster comparison was performed using ClusterProfiler R package (Yu G, OMICS: *A Journal of Integrative Biology*, 2012). Genes were uploaded in Enricher [60] and FunRich tool (version 3.1.1) [61] for KEGG pathway and Transcription factor enrichment of genes. 

### 4.6. Gene Interaction Network and Identification of Hub Genes

Differentially expressed genes (DEGs) with log ratio cutoff ≥ 2 were analyzed for gene interaction with GeneMANIA [62] in Cytoscape (V3.9.1) [63]. Important gene interaction network cluster modules were identified using Molecular Complex Detection (MCODE) Cytoscape plugin [64] sorted at a degree cutoff-2.0, node cutoff- 0.2, K-score-2, and max depth-100. Clusters were calculated for degree scores of nodes and genes with highest degree scores in cluster modules were identified as hub genes by Cytohubba plug-in on Cytoscape [65]. In clusters, genes function as nodes, and nodes with higher interactions with other genes unequivocally play crucial roles in cluster formation and are called HUB genes [66].

## 5. Conclusions

Our study provides an entry point for the identification of claudin-6 as a new prognostic marker for the evaluation and prognosis of gastric tumors

## Figures and Tables

**Figure 1 ijms-23-13977-f001:**
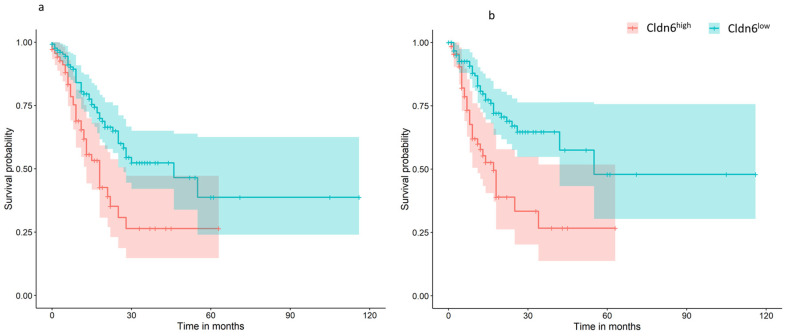
Cldn6 expression and patient survival. (**a**) Overall survival; (**b**) progression-free survival.

**Figure 2 ijms-23-13977-f002:**
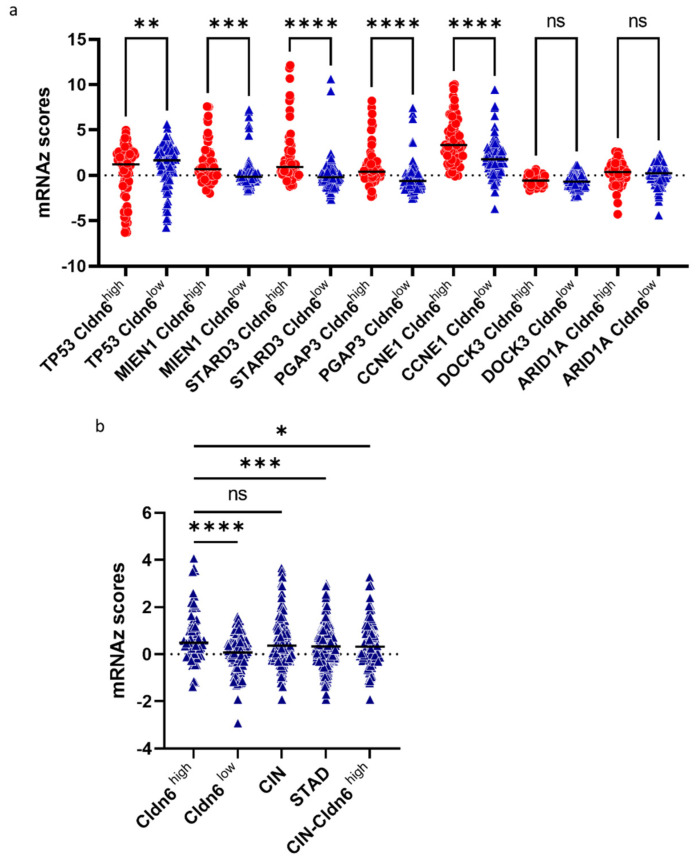
Significant genomic alterations between Cldn6^high^ and Cldn6^low^ groups. (**a**) mRNA expression z scores of TP53, MIEN1, STARD3, PGAP3, CCNE1, AIRD1A, and DOCK3; (**b**) ERBB2 expression differences between gastric cancer subsets. CIN = chromosomal instable; STAD = stomach adenocarcinoma; CIN−Cldn6^high^ = chromosomal instable subtype without the Cldn6^high^ subset. *p*-value * < 0.0255, ** < 0.0045, *** < 0.0002, **** < 0.0001.

**Figure 3 ijms-23-13977-f003:**
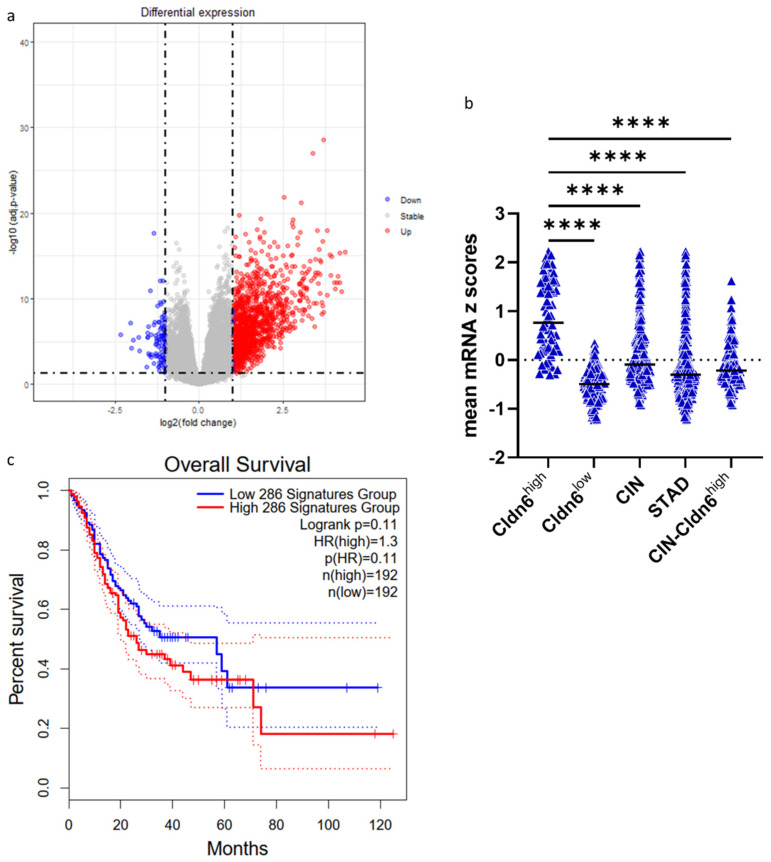
(**a**) Volcano plot representing upregulated and downregulated genes in Cldn6^high^ GC; (**b**) mean z scores of the genes with log ratio cut off ≥ 2 in the Cldn6^high^ subset in contrast to Cldn6^low^, CIN, STAD, and CIN−Cldn6^high^; (**c**) Survival analysis of GC patients with DEGs log ratio cut off ≥ 2 (signature groups). CIN = chromosomal instable; STAD = stomach adenocarcinoma; CIN−Cldn6^high^ = chromosomal instable subtype without the Cldn6^high^ subset. Statistical significance was evaluated with one-way ANOVA, adjusted *p*-Value, **** < 0.0001.

**Figure 4 ijms-23-13977-f004:**
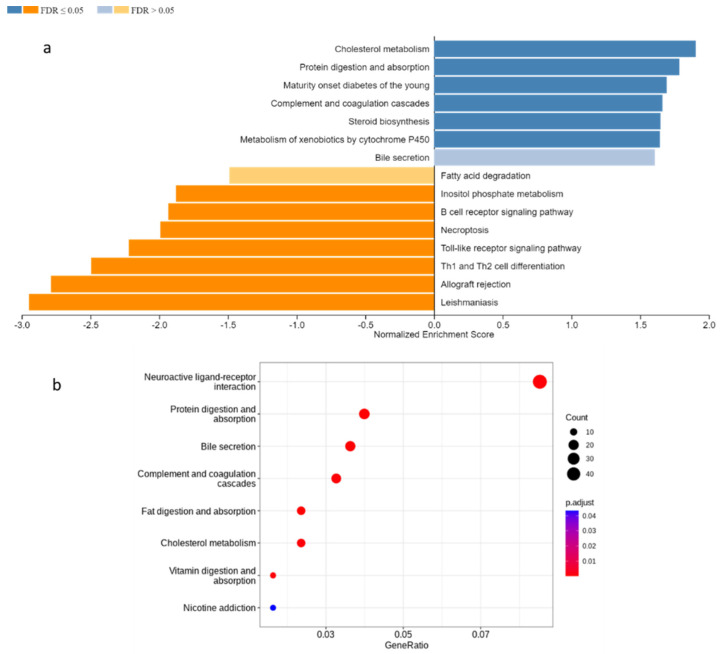
Gene enrichment analysis. (**a**) Gene set enrichment analysis of all DEGs using WEB-based GEne SeT Analysis Toolkit (WebGestalt, http://www.webgestalt.org/) (accessed on 27 March 2022). The normalized enrichment scores (NES) of top enriched (blue bars) and top depleted (orange bars) pathways are listed; (**b**) gene cluster comparison using DEGs with log ratio cutoff ± 1 (ClusterProfiler package in R). Color and size of dots represent the *p*-value and number of enriched genes in pathways sequentially.

**Figure 5 ijms-23-13977-f005:**
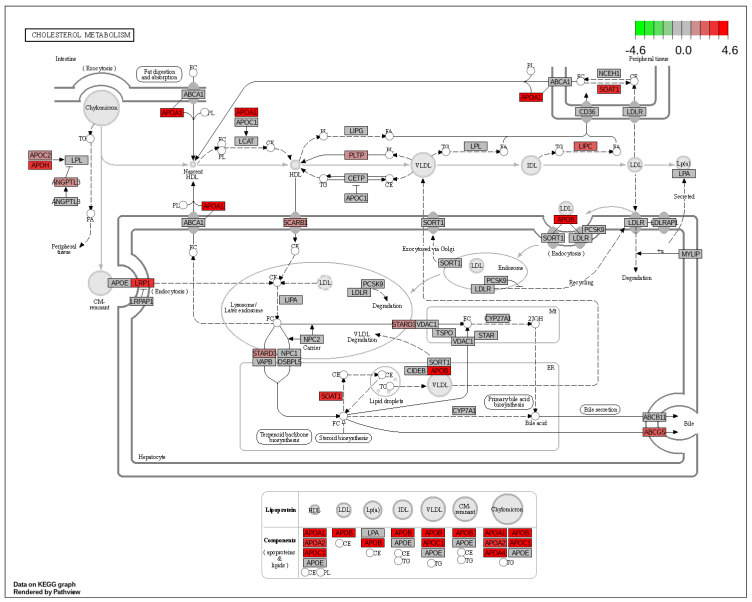
KEGG map of cholesterol metabolism in the Cldn6^high^ group. Each box represents a gene and the box color (green for downregulation, grey for no effect, red for upregulation) represents the gene inferring impact on the pathway.

**Figure 6 ijms-23-13977-f006:**
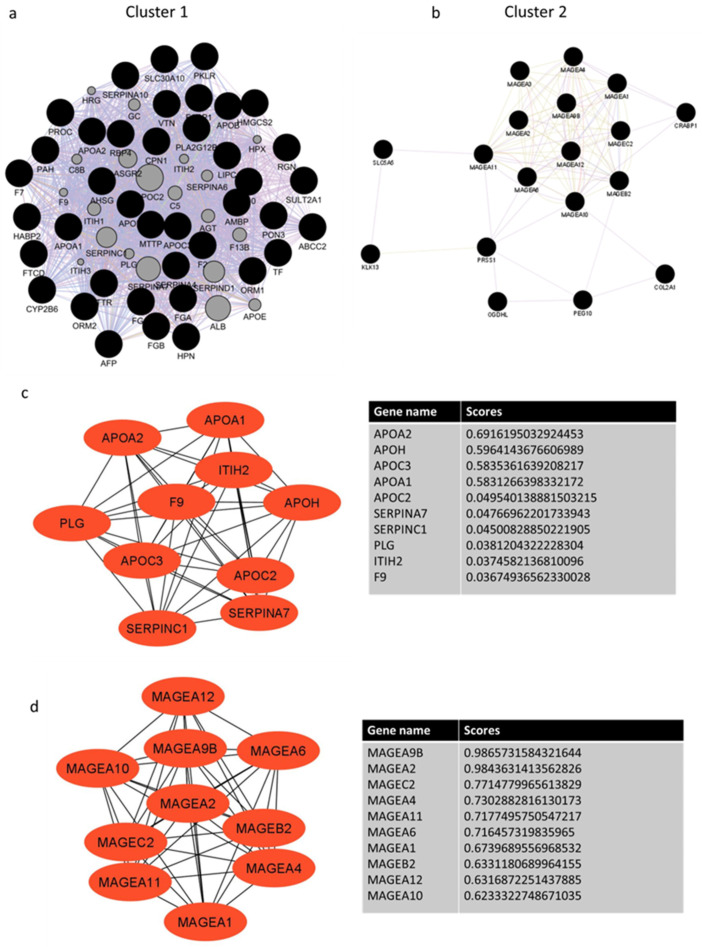
Gene interaction network of differentially expressed genes (DEGs) with a log ratio cutoff ≥ 2 for the identification of Hub genes. (**a**) Genes with a 54.44 score are included in cluster 1; (**b**) genes with an 8.3 score are included in cluster 2. The size of the circle represents the number of interactions/NODE, whereas the black color of the circle represents query DEGs whereas grey circles represent associated genes introduced by GeneMania; (**c**) represents the top 10 gene interactions with the highest degree scores in cluster 1 and (**d**) in cluster 2.

**Figure 7 ijms-23-13977-f007:**
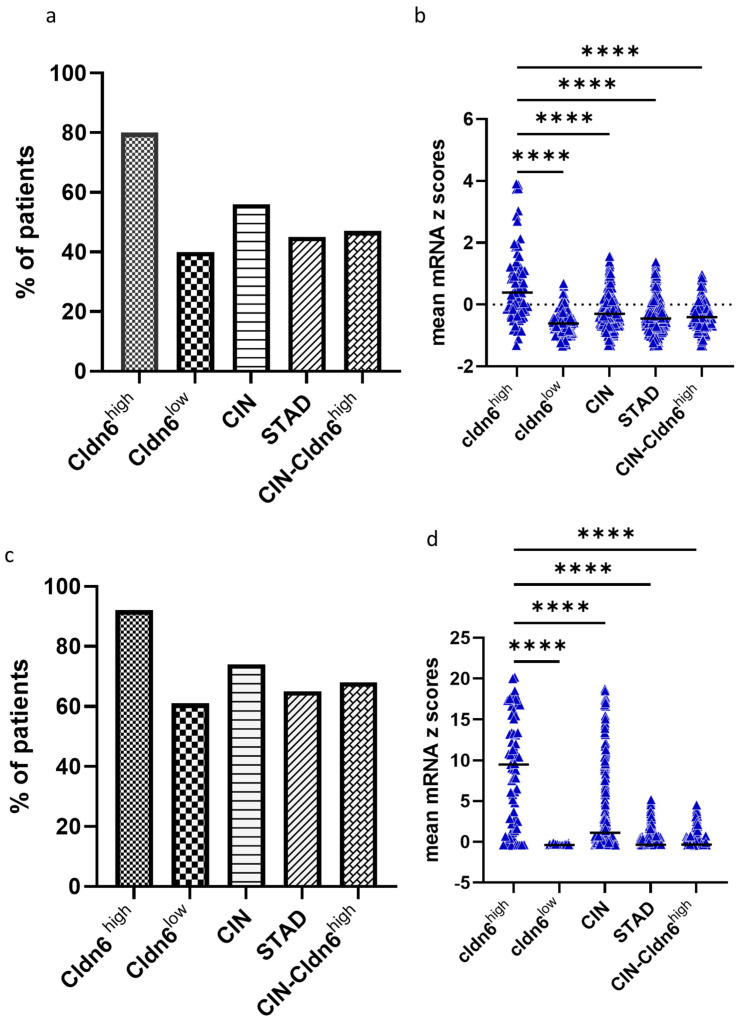
The percentage of gastric cancer patients expressing top 10 clusters 1 and 2 genes. (**a**) Alteration event frequency of top 10 cluster 1 genes among GC subsets and STAD; (**b**) mean z scores of top 10 cluster 1 genes in the Cldn6^high^ subset in contrast to Cldn6^low^, CIN, STAD, and CIN−Cldn6^high^ GC; (**c**) Alteration event frequency of top 10 cluster 2 genes among GC subsets and STAD; (**d**) mean z scores of top 10 cluster 2 genes in the Cldn6^high^ subset in contrast to Cldn6^low^, CIN, STAD, and CIN−Cldn6^high^ GC. CIN = chromosomal instable; STAD = stomach adenocarcinoma; CIN−Cldn6^high^ = chromosomal instable subtype without the Cldn6^high^ subset. Statistical significance was confirmed with one-way ANOVA, adjusted *p*-Value **** < 0.0001.

**Figure 8 ijms-23-13977-f008:**
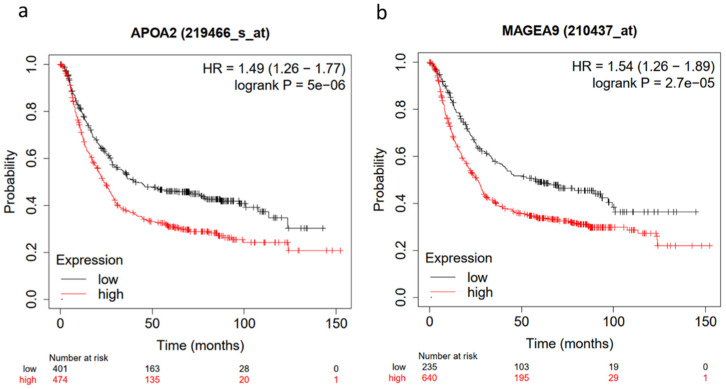
Survival analysis in GC tumors (n = 875). Kaplan–Meier survival curves: gastric adenocarcinoma with (**a**) low (n = 401) and high (n = 474) APOA2 expression (*p* = 5 × 10^−6^); and (**b**) low (n = 235) and high (n = 640) MAGE9 expression (*p* = 2.7 × 10^−5^).

**Figure 9 ijms-23-13977-f009:**
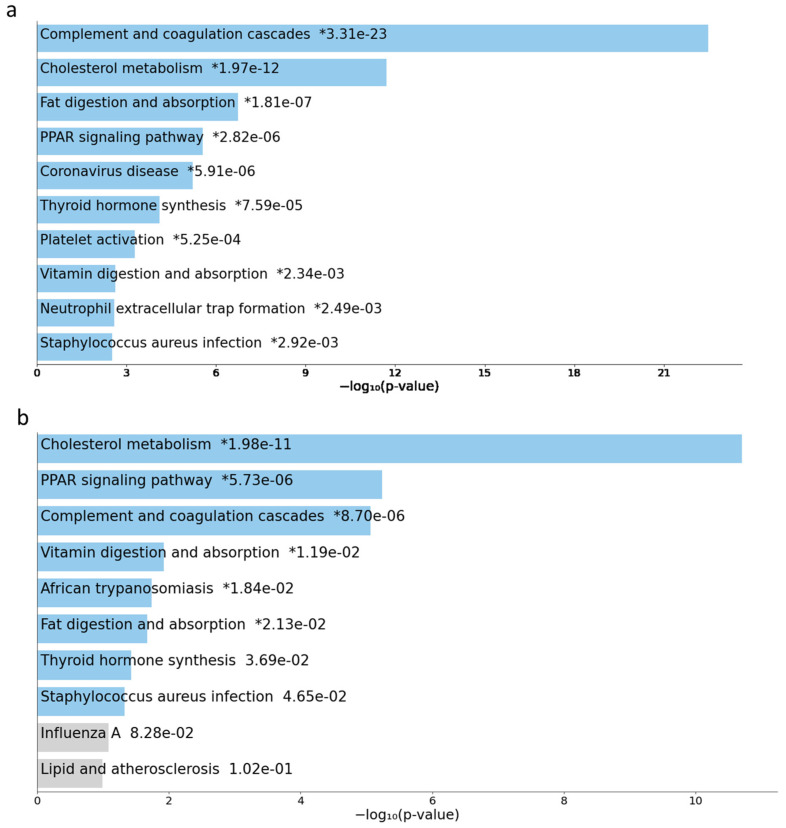
KEGG 2021 human (human cell signaling and metabolic pathways manually curated database, 2021 version) analysis of pathway enrichment performed using Enrichr on DEGs on cluster 1 genes; (**a**) all genes; (**b**) top 10 expressed genes. The horizontal axis represents the number of genes, and the y-axis represents the pathway names. *; p values derived from fisher exact text.

**Figure 10 ijms-23-13977-f010:**
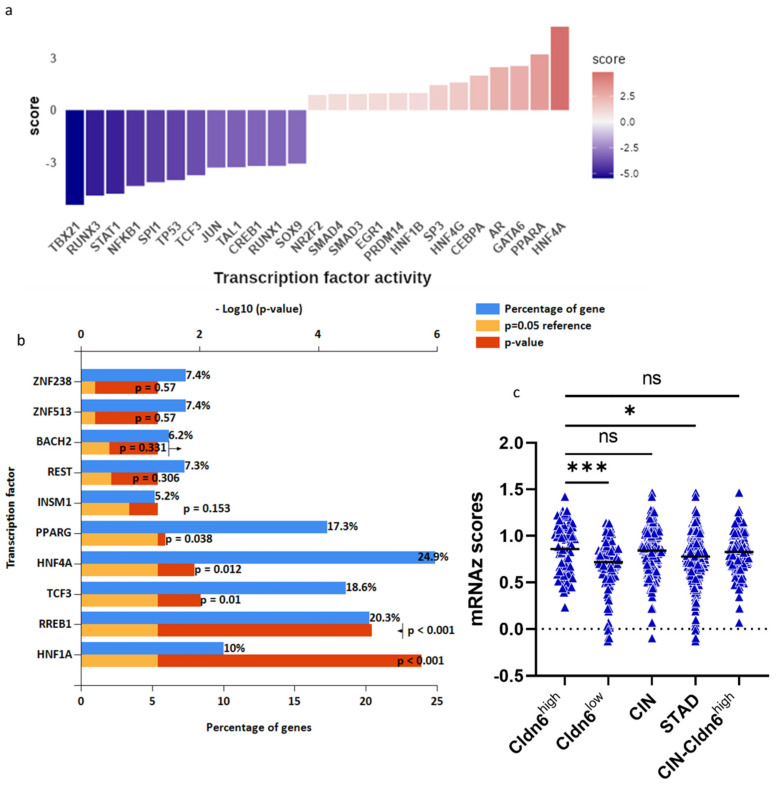
Transcription factors enrichment. (**a**) Normalized enrichment scores from the Dorothea database to infer transcription factor activity from differentially expressed genes; (**b**) using the FunRich database, blue bars represent the percentage of genes assigned to the indicated transcription factor, yellow bars show the reference *p*-value (0.05), and red bars show the calculated p-value of enrichment for the indicated transcription factor; (**c**) expression of HNF-4α among different GC groups. CIN = chromosomal instable; STAD = stomach adenocarcinoma; CIN−Cldn6^high^ = chromosomal instable subtype without the Cldn6^high^ subset. Statistical significance was confirmed with one-way ANOVA, adjusted *p*-Value, * < 0.0255, *** < 0.0002.

**Figure 11 ijms-23-13977-f011:**
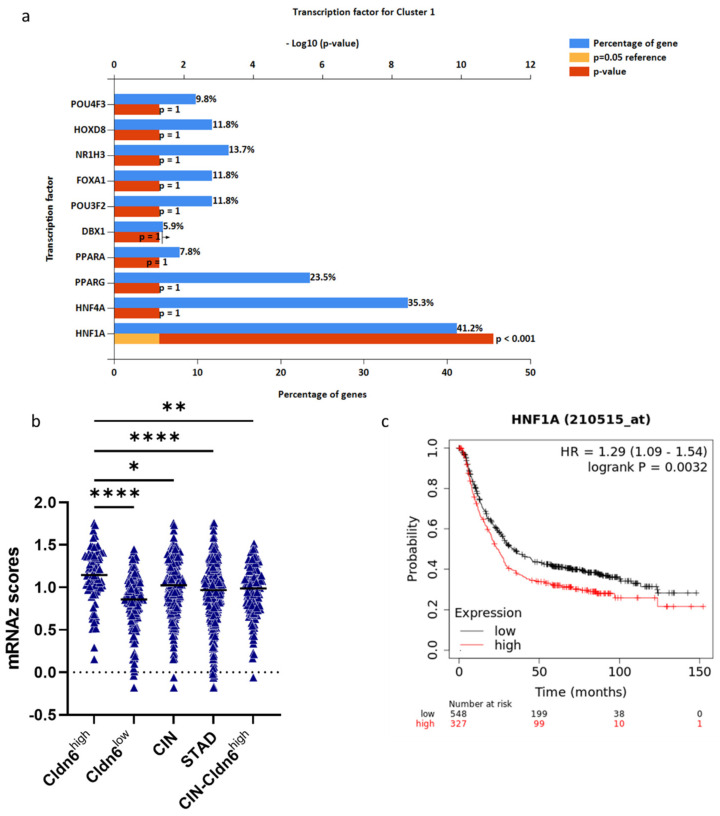
Transcription Factors enrichment of cluster 1 top 10 hub genes. (**a**) Transcription factors enrichment of top 10 hub genes of cluster 1 using the FunRich database. Blue bars represent the percentage of genes assigned to the indicated transcription factor, yellow bars show the reference *p*-value (0.05), and red bars show the calculated *p*-value of enrichment for the indicated transcription factor; (**b**) correlation of HNF-1α expression and patient survival in GC (n = 875); (**c**) expression of HNF-1α among different GC groups. CIN = chromosomal instable; STAD = stomach adenocarcinoma; CIN−Cldn6^high^ = chromosomal instable subtype without the Cldn6^high^ subset. Statistical significance was confirmed with one-way ANOVA, adjusted *p*-Value, * < 0.0255, ** < 0.0045, **** < 0.0001.

**Table 1 ijms-23-13977-t001:** Sorting of Cldn6^high^ and Cldn6^low^ tumors by TCGA molecular subtype.

Type	Cldn6^high^	Cldn6^low^
CIN	69	57
EBV	0	22
GS	2	18
MSI	0	38
POLE	0	3

CIN = chromosomal instable; EBV = Epstein–Barr virus positive; GS = genomically stable; MSI = microsatellite unstable; POLE = polymerase epsilon. The results are expressed as the number of patients for each subtype.

**Table 2 ijms-23-13977-t002:** Major genomic alterations between Cldn6^high^ and Cldn6^low^ groups.

Gene	Cldn6 ^high^ (n = 71)	Cldn6 ^low^ (n = 138)	Log Ratio	*p*-Value	q-Value
TP53	76.06%	31.88%	1.25	9.51 × 10^−10^	2.062 × 10^−5^
MIEN1	29.58%	5.80%	2.35	5.917 × 10^−6^	0.0321
STARD3	29.58%	6.52%	2.18	1.454 × 10^−5^	0.0487
PGAP3	28.17%	5.80%	2.28	1.536 × 10^−5^	0.0487
CCNE1	26.76%	5.07%	2.40	1.571 × 10^−5^	0.0487
DOCK3	0.00%	20.29%	<−10	3.143 × 10^−6^	0.0266
ARID1A	9.86%	39.13%	−1.99	3.682 × 10^−6^	0.0266

**Table 3 ijms-23-13977-t003:** Major differentially expressed genes in Cldn6^high^ gastric cancers.

Gene	Log Ratio	*p*-Value	q-Value
CLDN6	8.49	7.14 × 10^−41^	1.35 × 10^−36^
MAGEA6	6.12	1.62 × 10^−21^	4.36 × 10^−18^
IGF2BP1	6.12	3.63 × 10^−24^	1.71 × 10^−20^
MAGEA3	5.86	4.58 × 10^−21^	1.08 × 10^−17^
MAGEA4	5.62	2.11 × 10^−15^	6.33 × 10^−13^
APOA1	5.55	7.33 × 10^−18^	5.13 × 10^−15^
MAGEA2	5.37	1.62 × 10^−19^	2.35 × 10^−16^
MAGEA9B	5.04	1.11 × 10^−16^	5.00 × 10^−14^
MAGEA12	4.93	2.67 × 10^−17^	1.60 × 10^−14^
CSAG1	4.35	3.83 × 10^−16^	1.54 × 10^−13^
COL2A1	4.25	1.55 × 10^−11^	1.20 × 10^−09^
APOA4	4.2	1.29 × 10^−12^	1.44 × 10^−10^
PRAME	4.18	4.49 × 10^−16^	1.73 × 10^−13^
MAGEA10	4.14	2.63 × 10^−13^	3.73 × 10^−11^
TF	4.11	1.48 × 10^−12^	1.61 × 10^−10^
FGB	4.06	7.51 × 10^−12^	6.30 × 10^−10^
TTR	4.02	1.24 × 10^−12^	1.40 × 10^−10^
APOC3	3.96	1.33 × 10^−15^	4.24 × 10^−13^
DSCR8	3.91	1.82 × 10^−13^	2.77 × 10^−11^
LGALS7B	3.89	8.11 × 10^−16^	2.94 × 10^−13^

## Data Availability

The analysis was performed using the TGA STAD Pan-Cancer Atlas gene expression data.

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
