# Peer review of "Chromosomally Unstable Gastric Cancers Overexpressing Claudin-6 Disclose Cross-Talk between HNF1A and HNF4A, and Upregulated Cholesterol Metabolism"

_ijms, 2022, doi:10.3390/ijms232213977_

Round 1
Reviewer 1 Report
In this manuscript, by Dwivedi et al., the authors explored abnormal gene expression in claudin-6 increased gastric cancers to search for potential prognostic marker/therapeutic target by ways of statistical analysis.
This work might contribute to a better understanding of correlated gene expression in Cldn6high gastric cancers, however with nonnegligible problems in results presentation:
1. In the abstract, the authors claimed “inhibition HNF-4α and HNF-1α inhibition can be explored as therapeutic targets.” Except from the grammatical problem here with repeated “inhibition”, the current work might suggest a correlated higher expression of HNF-4α and HNF-1α in Cldn6high gastric cancers than its Cldn6low counterparts, however, it’s far from supporting “HNF-4α and HNF-1α inhibition to be explored as therapeutic target” without any preclinical and clinical studies.
2. The citations are not in correct order and jumped from [16] to [26], while citations in section 4. Materials and Methods starts from [17].
3. In table 1, looks like the authors are presenting the data in percentage, but without mentioning it anywhere. The data is also very confusing, does 69 takes 96.88% of 71, and 2 takes 3.13% of 71? Cldn6low data is also incorrect if it’s in percentage, the sum number does not make 100%, please double check and correct them accordingly.
4. Line 67, PLOE should be POLE, and abbreviations should be defined at first use. There are same problems elsewhere, e.g., “DEGs” in line 100 and “FDR” in line 126, please check through the manuscript and make the corrections.
5. Line 84, it’s not proper to claim “TP53 mRNA levels were downregulated in the Cldn6high subset”, as it’s actually “TP53 showed a slightly lower mRNA levels in the Cldn6high subset compared to Cldn6low subset”; Similarly in Abstract, the authors are not supposed to make misleading statements, like in line 17, it should be “Cldn6high expression coincided with higher mutations in TP53, MIEN1, STARD3, PGAP3, and CCNE1 genes compared to Cldn6low expression.”
6. The terms are not unified, like it’s jumping from “HNF-4α” to “HNF-4a”(line 172,340).
7. Fig 5 are mostly unreadable with the small fonts; Fig 6, there should be figure captions for a,b,c,d respectively. Similarly for Fig 8 caption, a and b should be indicated respectively.
8. What does "KEGG 2021 human" in Fig 9 indicate? In general, the figs are not clearly explained/presented in the main text.
9. Introduction of “Hub genes” from line 332 to line 337 should belong to section 1. Introduction.
Author Response
Enclosed you will find the revised manuscript entitled “Chromosomally Unstable Gastric Cancers Overexpressing Claudin 6 Disclose Cross-talk Between HNF1A and HNF4A, And Upregulated Cholesterol Metabolism” by Sanyog Dwivedi et al., which I hope that now is acceptable for publication.
All the comments were considered and the manuscript has now been modified accordingly.
Reviewer 1.
- In the abstract, the authors claimed “inhibition HNF-4α and HNF-1α inhibition can be explored as therapeutic targets.” Except from the grammatical problem here with repeated “inhibition”, the current work might suggest a correlated higher expression of HNF-4α and HNF-1α in Cldn6high gastric cancers than its Cldn6low counterparts, however, it’s far from supporting “HNF-4α and HNF-1α inhibition to be explored as therapeutic target” without any preclinical and clinical studies.
- We have modified the conclusion paragraph in the abstract and we completely agree that none of our results support the original paragraph. We have eliminated the idea of HNFs being used as therapeutic targets.
- The citations are not in correct order and jumped from [16] to [26], while citations in section 4. Materials and Methods starts from [17].
- The numbering of the citations has been ordered.
- In table 1, looks like the authors are presenting the data in percentage, but without mentioning it anywhere. The data is also very confusing, does 69 takes 96.88% of 71, and 2 takes 3.13% of 71? Cldn6low data is also incorrect if it’s in percentage, the sum number does not make 100%, please double check and correct them accordingly.
- We have modified the table and in its new form we only report absolute numbers of patients in each gastric cancer molecular subtype.
- Line 67, PLOE should be POLE, and abbreviations should be defined at first use. There are same problems elsewhere, e.g., “DEGs” in line 100 and “FDR” in line 126, please check through the manuscript and make the corrections.
- We have checked the manuscript and make sure that the POLE, DEG and FDR abbreviations are correctly defined. We have also asked an English native speaker to check the language use.
- Line 84, it’s not proper to claim “TP53 mRNA levels were downregulated in the Cldn6high subset”, as it’s actually “TP53 showed a slightly lower mRNA levels in the Cldn6high subset compared to Cldn6low subset”; Similarly in Abstract, the authors are not supposed to make misleading statements, like in line 17, it should be “Cldn6high expression coincided with higher mutations in TP53, MIEN1, STARD3, PGAP3, and CCNE1 genes compared to Cldn6low expression.”
- We have modified the text in the abstract and results section in order to clarify that TP53 increased slightly in Cldn6high in comparison to Cldn6low groups. This type of bioinformatic analysis although very helpful will always need a clinical confirmation, which we are performing.
- The terms are not unified, like it’s jumping from “HNF-4α” to “HNF-4a” (line 172,340).
- These terms and others have been unified throughout the manuscript
- Fig 5 are mostly unreadable with the small fonts; Fig 6, there should be figure captions for a,b,c,d respectively. Similarly for Fig 8 caption, a and b should be indicated respectively.
- The captions of all the figures have been re-written so we now believe that they are clearer. Because figure 5 is derived from an analysis software it is difficult to modify nevertheless we changed it so every gene in the pathway is now in capital letters and hopefully this will make it easier for your readers.
- What does "KEGG 2021 human" in Fig 9 indicate? In general, the figs are not clearly explained/presented in the main text.
- We have modified the heading of the figure and eliminated the KEGG 2021 human but we have also modified the figure foot note to explain what KEGG 2021 human means. We believe that this new revised version of the manuscript explain more clearly all the manuscript figures.
- Introduction of “Hub genes” from line 332 to line 337 should belong to section 1. Introduction.
- We have modified the abstract and the discussion section where Hub genes are mentioned and so a clearer concept of what a Hub gene is can be considered by the readers.
Reviewer 2 Report
The manuscript by Dwivedi et al. presented the analysis on how the high expression of Claudin-6 correlates with a more aggressive phenotype in chromosomally unstable gastric cancers. The authors found cholesterol metabolism was upregulated in Claudin-6 high gastric cancers and identified the hub genes. Although the analysis revealed some correlation between the high expression of Claudin-6 with some genes, I find it lack of validation.
Major points:
In section 2.2, the authors analyzed the genomic alternation in some genes. But they didn’t show how these genes were selected
There is some error for the Fig 2 legend.
Author Response
Enclosed you will find the revised manuscript entitled “Chromosomally Unstable Gastric Cancers Overexpressing Claudin 6 Disclose Cross-talk Between HNF1A and HNF4A, And Upregulated Cholesterol Metabolism” by Sanyog Dwivedi et al., which I hope that now is acceptable for publication.
All the comments were considered and the manuscript has now been modified accordingly.
Reviewer 2.
- The manuscript by Dwivedi et al. presented the analysis on how the high expression of Claudin-6 correlates with a more aggressive phenotype in chromosomally unstable gastric cancers. The authors found cholesterol metabolism was upregulated in Claudin-6 high gastric cancers and identified the hub genes. Although the analysis revealed some correlation between the high expression of Claudin-6 with some genes, I find it lack of validation. Answer: We have included in the relevant materials and methods section a brief explanation of how the genes we used to our analysis were validated. It is clear that the expression of thousands of different genes is modified in cancer so mathematical rules have to be utilized to start analyzing the most clinically relevant genes; we used genes that were significantly overexpressed.
- In section 2.2, the authors analyzed the genomic alternation in some genes. But they didn’t show how these genes were selected. Answer: As I have mentioned in the above paragraph the method we used to select the genes we analyzed was based on their significant overexpression.
- There is some error for the Fig 2 legend. Answer: As already mentioned, all the figures footnotes have been corrected.
Round 2
Reviewer 2 Report
The manuscript still lacks experiment validation. Without any kinds of experiment validation, the research makes very little sense.
All the analysis in this manuscript is correlation analysis. The authors should put some effort to understand how the overexpression of Claudin-6 leads to upregulated cholesterol metabolism